# Mental health among healthcare workers and other vulnerable groups during the COVID-19 pandemic and other coronavirus outbreaks: A rapid systematic review

Eleonora P. Uphoff[1], Chiara Lombardo[2,3], Gordon Johnston[2], Lauren Weeks[2], Mark Rodgers[4]*, Sarah Dawson[5], Catherine Seymour[2], Antonis A. Kousoulis[2], Rachel Churchill[1]

**1** Cochrane Common Mental Disorders, Centre for Reviews and Dissemination, University of York, York, United Kingdom, **2** Mental Health Foundation, London, United Kingdom, **3** Department of Public Health and Primary Care, University of Cambridge, Cambridge, United Kingdom, **4** Centre for Reviews and Dissemination, University of York, York, United Kingdom, **5** Population Health Sciences, Bristol Medical School, University of Bristol, Bristol, United Kingdom

* Mark.rodgers@york.ac.uk

**Data Availability Statement:** All relevant data are within the manuscript and its S1–S3 Appendices files.

## Abstract

### Introduction

Although most countries and healthcare systems worldwide have been affected by the COVID-19 pandemic, some groups of the population may be more vulnerable to detrimental effects of the pandemic on mental health than others. The aim of this systematic review was to synthesise evidence currently available from systematic reviews on the impact of COVID-19 and other coronavirus outbreaks on mental health for groups of the population thought to be at increased risk of detrimental mental health impacts.

### Materials and methods

We conducted a systematic review of reviews on adults and children residing in a country affected by a coronavirus outbreak and belonging to a group considered to be at risk of experiencing mental health inequalities. Data were collected on symptoms or diagnoses of any mental health condition, quality of life, suicide or attempted suicide. The protocol for this systematic review was registered in the online PROSPERO database prior to commencing the review (https://www.crd.york.ac.uk/prospero/display_record.php?RecordID=194264).

### Results

We included 25 systematic reviews. Most reviews included primary studies of hospital workers from multiple countries. Reviews reported variable estimates for the burden of symptoms of mental health problems among acute healthcare workers, COVID-19 patients with physical comorbidities, and children and adolescents. No evaluations of interventions were identified. Risk- and protective factors, mostly for healthcare workers, showed the importance of personal factors, the work environment, and social networks for mental health.

**Funding:** This work was supported by funding from the Mental Health Foundation and by the National Institute for Health Research (NIHR) through Cochrane Infrastructure funding to the Common Mental Disorders Cochrane Review Group (award no. NIHR129457). The views and opinions expressed herein are those of the review authors and do not necessarily reflect those of the NIHR, National Health Service (NHS), or the Department of Health and Social Care.

**Competing interests:** The authors have declared that no competing interests exist.

## Conclusions

This review of reviews based on primary studies conducted in the early months of the COVID-19 pandemic shows a lack of evidence on mental health interventions and mental health impacts on vulnerable groups in the population.

## Introduction

Since the start of the COVID-19 pandemic in China in December 2019, most countries and healthcare systems globally have been affected by the outbreak. As of 13 April 2021, nearly 3 million people are estimated to have died of COVID-19 and 137 million cases have been reported, with over 4.4 million cases and 127,346 deaths in the United Kingdom (https://coronavirus.jhu.edu/map.html). Billions of people have experienced impacts of the pandemic on their daily lives, with potential consequences for their mental health.

Factors mentioned as contributors to emotional distress include the restrictions imposed by governments around the world to try and slow the spread of the virus, the uncertainty about the course of the pandemic in future months and years, worries about getting infected and falling ill, financial insecurity and job loss, and social isolation [1].

Some groups may be more likely to experience mental health difficulties because of the pandemic than others [2, 3]. This includes groups generally thought of as more vulnerable due to their increased exposure to adverse circumstances and environments. Healthcare workers are more likely to be exposed to patients with COVID-19. People on a low income, low level of education, or low-wage job are at increased risk of job insecurity and unemployment and more likely to experience overcrowding, poor quality housing, and physical health conditions [2, 4]. Socially excluded groups such as prisoners, homeless, and refugees and asylum seekers may be less able to protect themselves from infection with COVID-19. There may be additional risks for people of minority and ethnic groups, pregnant women, and young adults and children [2]. Adults belonging to a minority ethnic group are for example are more likely to be 'key workers' [5] and more likely to experience overcrowding, poverty, and insecure employment [6]. They may also be less likely to receive a diagnosis or treatment for mental health conditions [7]. Those with physical health conditions have a worse prognosis than others [8], and people with existing mental health conditions may find symptoms are exacerbated during the pandemic [2, 9].

Primary research on COVID-19 is evolving rapidly and the International Prospective Register of Systematic Reviews (PROSPERO) counts 398 registered systematic review titles on mental health and COVID-19 as of 13 April 2020 (https://www.crd.york.ac.uk/prospero). A living systematic map of the evidence counts 3013 records relating to mental health impacts of COVID-19 as of 13 April 2021 (EPPI-Mapper 2020) [10].

Our rapid systematic review synthesises evidence currently available from systematic reviews on the impact of COVID-19 and other coronavirus outbreaks on mental health for groups of the population thought to be at an increased risk of detrimental mental health impacts.

## Methods

This systematic review was conducted in a short timeframe, to inform mental health policy and practice in the UK during the COVID-19 pandemic. We used rapid review methodology

based on guidance from the Cochrane Rapid Review Methods Group, to ensure a pragmatic yet high quality systematic review. At the time of conducting our study, we were not aware of another review of systematic reviews on this topic.

The review is reported according to the Preferred Reporting Items for Systematic Reviews and Meta-Analyses (PRISMA) guidelines [11]. The PRISMA checklist can be found in the S1 Appendix.

## Literature searches

We searched MEDLINE (2002 onwards), the Rayyan CORD-19 database specific to research on COVID-19 (https://www.semanticscholar.org/cord19), and two preprint databases MedRXiv (www.medrxiv.org) and PsyArxiv (www.psyarxiv.com). The full search strategy can be found in the S2 Appendix.

Search terms related to coronaviruses, mental health, systematic reviews, frontline workers, and inequality. The search was restricted to publications in English.

## Eligibility criteria

**Study design.**   Systematic reviews with or without meta-analyses were eligible for inclusion.

**Condition.**   To be included, a review had to focus on mental health, which may include positive mental health (well-being), any signs or symptoms of psychological distress and sub-threshold conditions (for example, mild depression), and mental illness including common mental disorders and severe mental illness. Reviews of people with a pre-existing mental health condition were eligible for inclusion, as long as determinants of inequality and impact on mental health were assessed.

**Population.**   Reviews of adults and children were eligible for inclusion, as long as they resided in a country where a coronavirus outbreak was taking place, and they belonged to a group considered to be at risk of experiencing mental health inequalities. These groups or determinants of inequality included: age (children and young people, older adults), multi-morbidity, learning difficulties, socially excluded groups (prisoners, homeless, refugees and asylum seekers), low income, financial insecurity, employment, education, social disadvantage, gender reassignment, sex, sexual orientation, race/ ethnicity, pregnancy and maternity, and religion [1].

Healthcare workers were considered to be a relevant group for this review, as they are likely to be at increased risk of poor mental health as a result of increased exposure to end of life care, moral injury, and increased risk of infection [1]. This includes frontline staff, emergency workers, and other staff working in a healthcare setting and/or those providing healthcare. We accepted any definition of healthcare workers as defined in the systematic reviews.

Exposure to a coronavirus outbreak was defined as residing in a country with 10 or more registered cases during the relevant time period: SARS-CoV-2 (global, 2020), SARS (SARS-CoV) (China, Hong Kong, Taiwan, Canada, Singapore, Vietnam, US, Philippines) (2002–2004), MERS (MERS-CoV) (Saudi Arabia, South Korea, United Arab Emirates, Jordan, Qatar) (2012–2020).

**Outcomes.**   Reviews were eligible for inclusion if they reported on one or more of the following outcomes: symptoms of any mental health condition (assessed using any measure including self-reported symptoms), diagnosis of any mental health condition, quality of life, suicide or attempted suicide.

## Study selection

De-duplicated records were uploaded in Covidence [12]. Titles and abstracts were screened by four reviewers in duplicate. Conflicts were resolved through discussions within the team of

reviewers, with a senior systematic review expert available for consultation. Full-text manuscripts were sought and screened in duplicate by the review team. At this stage, reasons for exclusion were recorded. Abstracts and titles for which no full-text manuscript could be obtained within two weeks of starting data extraction were excluded.

### Data extraction

Multiple records of the same review were linked before commencing data extraction in Covidence [12]. Data were extracted by one reviewer and checked by an experienced systematic reviewer on the author team. Disagreements were resolved through discussion within the author team.

Data were extracted on review methods, population, interventions and outcomes, and on the methodology, country setting, and number of included primary studies. For each review, we extracted the references of included primary studies to assess overlap between reviews.

Due to the rapid nature of this review, study authors were not contacted to request missing data.

### Quality assessments

Quality assessments were conducted by one of the four reviewers and checked by an experienced systematic reviewer on the author team. Disagreements were resolved through discussion. We used the CASP checklist for systematic reviews to critically appraise the quality of included systematic reviews at the study level [13]. The results of these assessments were used to inform our conclusions and implications for future research, for practice, and for patients and the public.

### Synthesis

The extracted data were used to produce a narrative synthesis of results by type of inequality group, as we expected the study designs and data to be too heterogeneous for meta-analyses to be appropriate. Quantitative outcome data were summarised in tables, including prevalence rates of mental health conditions and symptoms.

### Patient and public involvement

We used a novel partnership working approach to complete this review. Our core team contains researchers with academic, third sector and lived experience, bringing a range of different skills and perspectives to the project. All members participated as equals, with tasks agreed and progress monitored in weekly team meetings.

In order to contextualise the results into a UK setting and to inform implications for practice, we conducted a two-hour consultation event with a group of healthcare workers employed both in the community and in the acute sector to discuss findings of the review. The consultation group consisted of six healthcare workers based in hospitals or community settings in the UK, including a project manager in women and children's services, a pharmacist, a trainee/ assistant clinical psychologist, a paediatric speech and language therapist, and two mental health nurses working with COVID-19 patients. All outputs of this work will be shared with the participants.

## Results

### Study selection process

The MEDLINE database was searched on 8 and 9 July 2020, in addition to the CORD-19 database specific to research on COVID-19 (10 August 2020), and two preprint databases MedR-Xiv (11 August 2020) and PsyArxiv (22 July 2020).

After removing duplicates, 746 records were screened. Twenty-five reviews were included in the synthesis (Fig 1). Most common reasons for exclusion were that the review did not assess a vulnerable group or determinant of inequality (N = 9), that the study was not a systematic review (N = 9), and that the publication was not in English (N = 7).

The 25 included systematic reviews included 715 references to primary studies, of which 236 were unique references. The most cited reference was included in 14 reviews [14].

We have not examined primary studies in this review of reviews. It is therefore likely that not all primary studies contributed data to our study. For example, some primary studies may have reported on different infectious diseases. There may also be multiple records of the same study among these references, for example pre-print and peer-reviewed publications.

## Description of included reviews

Literature searches of the included reviews were conducted between the 10th of March 2020 and the 10th of July 2020. Two included records were systematic review protocols for which no results had been published yet [15, 16].

Most reviews reported on COVID-19 or a mix of coronavirus outbreaks, while one review conducted before the COVID-19 pandemic assessed mental health impacts of the SARS outbreak (Table 1) [17]. Nineteen out of 25 reviews focussed on healthcare workers, predominantly in a hospital setting.

Reviews included primary studies conducted in a wide range of countries: Australia, Bangladesh, Canada, China, France, Germany, Greece, Hong Kong, Hungary, India, Indonesia, Italy, Iran, Ireland, Israel, Japan, Liberia, Macau, Malaysia, Mexico, Netherlands, New Zealand, Nigeria, Republic of Congo, Saudi Arabia, Senegal, Sierra Leone, Singapore, South Korea, Spain, Sweden, Taiwan, Thailand, Turkey, Uganda, UK, and USA. Most studies were conducted in China, particularly those focussed on COVID-19 at the start of the pandemic. As a study setting, China was represented in all included reviews. Two reviews included primary studies from China only [18, 19].

Most reviews included a wide range of primary study designs (Table 1). Some specified eligibility criteria for study design, for example relating to the minimum sample size [20, 21], assessment of mental health problems [19, 22] or quality of reporting and study conduct [18]. Fourteen reviews excluded studies not conducted in English.

## Prevalence of symptoms and mental health conditions

Included reviews reported on general mental health or symptoms or diagnoses of mental health problems, including anxiety, depression, PTSD, acute stress disorder, burn-out, sleep problems or insomnia, psychological distress, emotional exhaustion, alcohol intake and substance abuse, adjustment disorder, grief, and eating disorders.

Prevalence rates provided in Table 2 reflect the average rates reported in the included systematic reviews. Primary studies estimated these prevalence rates using various measures and diagnostic criteria.

**Healthcare workers.** Multiple reviews included estimates of mental health problems or symptoms among healthcare workers during a coronavirus outbreak. Estimates varied from 12% for anxiety in one review of healthcare workers in hospital [23], to 51% for depression and PTSD in another review [24].

Healthcare workers may have a higher baseline risk of adverse mental health outcomes due to the nature of their work. However, there were some indications that mental health may be further affected as a result of working during an infectious disease outbreak. A year after the SARS outbreak, healthcare workers were six times more likely to experience psychiatric

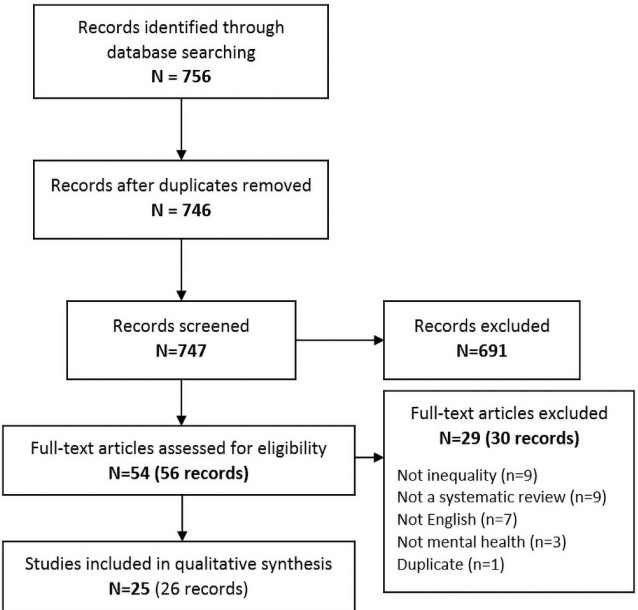

**Fig 1. Study selection process.**

**Table 1. Characteristics of included studies.**

| Study characteristic | Number of studies |
|---|---|
| Type of coronavirus | |
| COVID-19 only | 12 |
| Mix, including COVID-19, SARS, MERS, Ebola, influenza A/H1N1 | |
| ('swine flu'), influenza A/H7N9 ('avian influenza' or 'bird flu') | 12 |
| SARS only | 1 |
| Type of inequality aspect[a] | |
| Healthcare workers | 19 |
| Children and adolescents[b] | 5 |
| Patients with pre-existing conditions[b] | 2 |
| Homeless | 1 |
| Settings of included primary studies | |
| China only | 2 |
| Mix of countries | 21 |
| Designs of included primary studies | |
| Primary/ empirical studies, range of designs | 10 |
| Observational/ cross-sectional | 4 |
| Experimental or observational with control group | 1 |
| Qualitative studies | 1 |
| Any/ not specified | 9 |

[a] Some reviews included more than one group.

[b] One of these studies is a protocol.

**Table 2. Prevalence of mental health problems by equity group.**

| Population group | Mental health problem | Prevalence[a] |
|---|---|---|
| Healthcare workers | Anxiety | 12–45% |
| | Depression | 20–51% |
| | PTSD | 19–51% |
| | Psychological distress | 37% |
| | Acute stress disorder | 31% |
| | Burn-out | 29% |
| | Sleep problems | 34–37% |
| | Combination of mental health problems | 34% |
| COVID-19 patients with other health conditions (cancer, type 2 diabetes, Parkinson's) | Anxiety | 40–82% |
| | Depression | 50% |
| Children and adolescents[b] | Anxiety | 19–37% |
| | Depression | 35–44% |
| | PTSD | 6% |
| | Psychological distress | 40% |
| | Acute stress disorder | 17% |

[a] These estimates represent the range of estimates presented in included reviews, which includes pooled estimates from meta-analyses as well as estimates from primary studies reported in included reviews.
[b] Prevalence rates associated with quarantine and social isolation during infectious disease outbreaks.

symptoms than others [24]. One to two years after the SARS outbreak, 30% of healthcare workers with high levels of exposure to SARS patients still reported a high level of emotional exhaustion [24].

Quarantine was associated with acute stress disorder, PTSD symptoms, and alcohol intake among healthcare workers [17]. Insomnia was found to be higher in healthcare workers than the general population, although other adverse mental health outcomes showed similar rates between the groups [25].

**Patients with pre-existing conditions.** Reviews reported estimates of 40 to 82% for anxiety and 50% for depression among COVID-19 patients with pre-existing physical health conditions [26]. Two reviews reported that existing mental health problems including anxiety, may worsen as a result of exposure to a coronavirus pandemic [27, 28]. However, no comparison of prevalence rates was made.

**Children and adolescents.** For children and adolescents, quarantine of varying length and nature due to disease outbreaks was associated with a higher likelihood of developing acute stress disorder, adjustment disorder, symptoms of grief, and PTSD [29]. Different reviews reported estimates of 19 to 37% for anxiety, 35 to 44% for depression, 6% for PTSD, 40% for symptoms of psychological distress, and 17% for acute stress disorder [22, 30]. Reports of mental health symptoms in college students included anxiety, depression, substance abuse, sleeping disorders, and eating disorders [21].

Children with cystic fibrosis had lower levels of anxiety about the COVID-19 pandemic than children without cystic fibrosis, but their parents had increased levels of anxiety. However, for children with ADHD the pandemic worsened their symptoms [21].

## Risk and protective factors

Although many reviews reported risk factors and protective factors for the mental health of groups of the population, none of the reviews included studies evaluating interventions to improve mental health during and after a coronavirus outbreak. Risk- and protective factors were derived from a range of primary study methods including surveys and qualitative studies.

**Healthcare workers.** Table 3 lists risk factors and protective factors for healthcare workers identified among the included reviews. Findings on younger age as a risk factor for adverse mental health outcomes were mixed. Other risk factors, such as being a nurse and/ or female, working in a role with high risk of exposure, and experiencing stigma were frequently mentioned in the included reviews.

Table 4 lists factors identified as protective factors in the included reviews. Several of the protective factors corresponded with risk factors. For example, sense of control versus sense of loss of control, social support versus social isolation, and feeling unprepared versus training and education as well as experience in the job. Frequently mentioned factors included experience in the job and efficient guidelines and structures in hospitals to manage care for patients with COVID-19 and protect healthcare workers.

**Children and adolescents.** Children and adolescents may be at increased risk of experiencing adverse mental health outcomes when experiencing stigma, social change such as school closures, and changes in household interactions [29]. Risk factors related to the family and community included parental distress, financial strain, living in a high-risk area, and living in a rural area. Results for age and sex as risk factors were mixed [30].

Better awareness of COVID-19 along with media entertainment, reading, and physical activity may protect against the negative mental health impacts of COVID-19 in this group [21, 30].

**Table 3. Risk factors for adverse mental health outcomes.**

| Domain | Risk factors |
|---|---|
| Personal characteristics and circumstances | Female |
| | Younger age |
| | Lower household income |
| | Physical or previous mental health condition |
| | Being single |
| | Experiencing quarantine |
| | Worries about risk of getting infected |
| | Sense of loss of control |
| | Disruption to personal life |
| | Feeling unprepared |
| Work environment | Nurse |
| | High risk of contact with patients/ frontline worker |
| | Infected colleague |
| | Working in hardest hit area |
| | Job stress and dissatisfaction |
| | Precautionary measures perceived as impediment |
| | Non-voluntary assignment to high-risk role |
| Social network | Worries about family members getting infected |
| | Social rejection/stigma |
| | Social isolation |

**Table 4. Protective factors for adverse mental health outcomes.**

| Domain | Protective factors |
|---|---|
| Personal characteristics and circumstances | Sense of control |
| | Coping ability/resilience |
| | Experience in the job |
| | Sense of duty/sense of altruism |
| | Acceptance of risk |
| Work environment | Availability of medical resources |
| | Efficient healthcare system |
| | Infection control and precautionary measures in place |
| | Strict implementation of guidelines |
| | Availability of training and education |
| | Good communication/receiving up-to-date information |
| | Access to mental health support/psychological interventions |
| | Adequate time off work |
| | Balanced workload |
| | Working in a managerial or administrative role |
| | Peer support/having a cohesive team |
| Social network | Social support |

**Older people.**   Very limited evidence was available on risk factors for older people. One review stated that the increased risk of transmission of coronavirus and the increased risk of complications from COVID-19 may affect older people, particularly those with existing mental health symptoms [31].

**Homeless people.**   Only one review addressed the mental health of homeless people in relation to coronavirus, hypothesising that a higher exposure to coronavirus in this group may lead to negative mental health outcomes [31].

No relevant systematic review evidence was found for other groups that may be considered vulnerable in terms of mental health during a coronavirus outbreak.

## Quality assessment

Fig 2 shows a summary of results of the quality assessment per included review, per CASP item, for the 23 reviews reporting results. Items 6, requiring a summary of findings, was not completed as outcome data was extracted separately. Item 10, on harms and benefits of interventions, was not applicable to any of the included reviews as none of the reviews evaluated interventions.

**Selection criteria.**   All reviews except for one [31] were found to ask a specific research question and all but two seemed to include papers with study designs appropriate to the research question. Study selection criteria were unclear for one review [31] and included reviews with a systematic search, rather than systematic reviews, for another study [27].

**Literature search.**   Several reviews restricted their searches to a few online databases only, without including grey or unpublished literature, input from experts, reference list searches, or non-English literature. For 12 reviews, it was therefore not possible to know whether they were likely to identify all relevant primary studies.

**Quality assessment.**   Six reviews either did not include a quality assessment of primary studies or did not report whether quality assessment was performed. The authors of one review reported that a risk of bias assessment was conducted, but no results were reported [32].

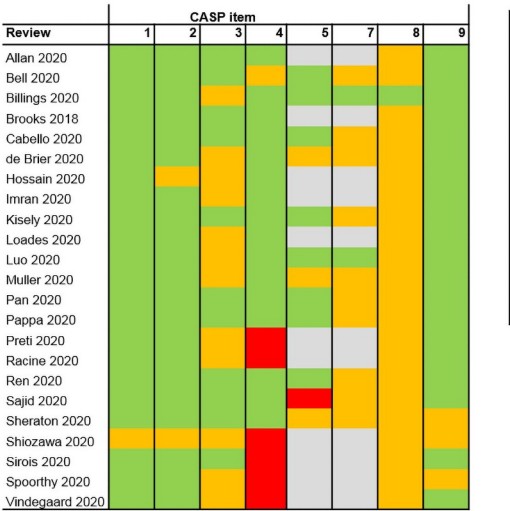
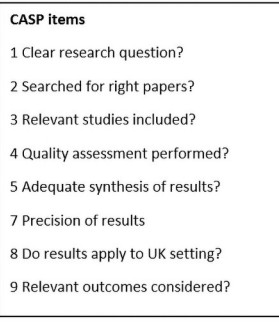

**Fig 2. Summary of CASP quality assessment of included reviews.** Light Green: High quality, Orange: Unclear or medium quality, Red: Low quality, Light Gray: Not applicable; no meta-analysis or combined effect estimates.

**Combining results.** Results combined in meta-analyses generally showed a lack of precision; confidence intervals were wide because primary studies included widely varying estimates. In part, this is likely to be a result of combining prevalence estimates obtained in different settings, using different methods and diagnostic criteria. For two reviews, it appeared various mental health outcomes were combined in one estimate [21, 25]. Apart from the estimates being imprecise, they are not likely to be meaningful.

**Applicability of results.** This review was conducted to inform policy in the United Kingdom for different groups of people. For most included reviews, most of the evidence was collected in China, reflecting the evidence base in the first few months of the COVID-19 pandemic as well as the epicentre of the SARS outbreak in 2002–2003. This makes it difficult to extrapolate findings to the UK setting. For example, different healthcare systems may lead to differences in mental health impacts on healthcare workers across countries and regions. Reviews of healthcare workers mostly included primary studies conducted in hospital settings, which may not be informative to healthcare in other settings such as nursing homes or community-based healthcare provision.

One review included qualitative studies covering a wide range of countries [33]. The authors considered differences and similarities in results from different studies and strong common themes emerged from the data. We concluded this review may reliably inform policy on COVID-19 for UK healthcare workers.

**Outcomes.** Most reviews considered outcomes relevant to mental health, including diagnoses of mental health conditions and symptoms of mental health difficulties. Three reviews were judged as 'unclear' for this item. For two reviews it was unclear which outcomes were eligible for inclusion in the review [31, 34]. Another review combined various outcomes in one meta-analysis; it was unclear which outcomes were combined and how this was done [25].

## Discussion

After synthesising data, preliminary findings were presented to a panel of healthcare workers to inform the synthesis and our discussion section including implications for policy and practice.

## Summary of findings

The 25 systematic reviews included in our rapid systematic review incorporated primary studies identified in the early stages of the COVID-19 pandemic. Most of the evidence was based on primary studies of coronavirus outbreaks in China and other countries outside of the UK, in hospital settings.

The burden of symptoms of mental health problems such as depression, anxiety, PTSD, distress, sleep problems, and burn-out appeared high among samples of healthcare workers, COVID-19 patients with physical comorbidities, and children and adolescents. As these studies generally did not include a control group or repeated measurements, it is not possible to know whether these symptom levels are higher than usual or whether symptoms increased over time.

None of the included reviews reported on evaluations of mental health interventions. Our inventory of risk- and protective factors, mostly for healthcare workers, demonstrates the perceived importance of practical and emotional support in the workplace, which in turn may influence personal characteristics of importance to mental health.

## Completeness of the evidence and relevance for UK healthcare

Other recently published reviews on hospital workers during a pandemic confirms that mental health problems such as symptoms of post-traumatic stress are common [35–37]. A systematic review based on studies published up to April 2020 found that symptoms of psychological distress and mental ill health were common during the first few months of the COVID-19 pandemic, but there was no reliable evidence to suggest that prevalence rates were higher among healthcare workers [37]. An August 2020 update of a living systematic review we included found that a higher compared to a lower risk of exposure among clinical staff is associated with an increase in anxiety and depression symptoms but not PTSD [36].

**Vulnerable groups and equity.**   The included systematic reviews were mostly based on primary studies from China and other Asian countries. Evidence was lacking for vulnerable groups other than healthcare workers. As noted by one of the panel members, even among healthcare workers there was little consideration in the literature of staff belonging to a minority ethnic group or those who identify as LGBTQ or have disabilities or physical health problems. One panel member raised the problematic practice of staff with disabilities being sent home under UK shielding guidance and therefore being exposed as having a disability.

**Hospital versus community setting.**   All of the included reviews focussed on healthcare workers in hospital. While the evidence on risk- and protective factors was mostly confirmed by our panel of healthcare workers, panel members raised important additional factors relevant to community healthcare workers.

Among our panel members working in community settings, the sudden change from working in the community to working from home affected the wellbeing of staff, particularly when personal circumstances and ability to work from home were not considered. Staff were also concerned they were not reaching disadvantaged groups in the community and delivering services remotely with inadequate resources caused stress.

Secondly, whilst panel members acknowledged they could promote their own mental health, for example by using social media to stay in touch with friends and family or by switching off from the continuous news cycle of COVID-19 related updates, the group identified support from colleagues as the most important form of support for their mental health. Hospital workers on the frontline emphasised the emotional support and sense of community that was present among colleagues working on the same wards. The panel agreed with the evidence on the importance of voluntary assignment to roles; staff who volunteered to work in these roles

were perceived to be strongly motivated by a sense of duty and a great willingness to help. At odds with the evidence, several panel members felt that frontline workers were coping well, while colleagues in roles with a lower risk of infection may have experienced more stress. Panel members experienced social stigma from colleagues not working with COVID-19 patients, rather than from friends and family. Community staff who had to work from home experienced more isolation and felt they were not fulfilling their sense of altruism and desire to help.

The importance of workplace support for mental health was evident from the literature as well as the panel discussions. Panel members valued psychological support, opportunities to talk, breaks, coffee mornings, training, offers of accommodation on hospital grounds, and a willingness from management to talk openly about wellbeing and recognise the importance of mental health. In line with the evidence from systematic reviews, timely communication, acting on feedback from staff, and clear guidance and expectations in the workplace were seen as important. Where this was not in place, for example for healthcare workers in the community who were told to work from home without the necessary support or resources in place, this was said to be a source of stress. In addition, redeployment not being used to its full potential meant those who were motivated to work missed the sense of fulfilment they derived from helping others. Among our panel members, those working on COVID-19 wards reported a clearer organisational structure and a greater sense of control compared to other hospital wards.

For those working in commercial settings, there may be additional pressure on workers to achieve the same level of profits for the company despite restrictions and reduced staffing levels. This was thought to come at the expense of wellbeing and safety of staff and customers.

Finally, among hospital and community workers in our panel, room for improvement was noted regarding practical support such as the limited availability of protective equipment at the start of the COVID-19 pandemic and the ongoing shortage of healthcare workers. Community workers required tools and resources that facilitate working remotely. Panellists also called for awareness regarding difficulties in reaching members of the community, for example for those who have limited or no access to the Internet.

**Social network.** In line with evidence from systematic reviews, panel members noted that family and friends could be a source of support as well as stress. For some, their social network largely consisted of healthcare workers, who were able to understand the situation in the workplace. Worries about spreading the illness to household members and family and friends being stressed about the risks of infection for healthcare workers were said to negatively impact on mental health.

**Government action.** Although included reviews touched on societal risk- and protective factors, they did not explicitly address the role of national governments. Our UK based panel members highlighted the need for practical, tangible support such as the availability of protective equipment and adequate pay. They unanimously reported a sense of uneasiness with the UK government focus on showing support through orchestrated initiatives such as the weekly 'Clap for Carers'. This "all fur coat and no knickers" approach was perceived to be a distraction from the lack of practical support provided (e.g. need for protective equipment, better resources, better pay). The problematic nature of the 'healthcare heroes' narrative has been questioned by others [38].

## Quality of evidence

We assessed the quality of included systematic reviews, rather than the quality of primary studies included in those reviews. Possibly due to the urgency of research on COVID-19, key

aspects of systematic review methodology such as comprehensive searches and quality assessments were lacking from several reviews. For all but one review we could not establish whether results would be applicable to the UK setting. Our panel of healthcare workers helped to put findings in the UK context, confirmed the importance of many of the identified risk- and protective factors for mental health, and put findings in context of healthcare in the UK in hospitals and in community settings.

## Limitations

Our rapid review has several limitations which may hinder the applicability of the evidence. We restricted our review to include English language publications only, which may mean we missed relevant reviews. We also restricted our selection criteria to include only systematic reviews, which means many recently published primary studies will have been missed.

This systematic review was conducted at a relatively early stage of the COVID-19 pandemic. Many primary studies and literature reviews will follow in months and years to come. A future update of this review of reviews would be able to incorporate a much larger and more diverse evidence base.

## Implications for research and practice

As the literature on COVID-19 grows, we hope it will evolve to include the perspectives of those groups at higher risk of experiencing adverse mental health impacts of the pandemic. High quality systematic reviews, incorporating the growing global literature from different countries and settings, are needed to inform mental health interventions that can mitigate negative impacts on mental health of vulnerable groups. This should include attention for those belonging to multiple vulnerable or minority groups, such as older people with physical health problems, ethnic minorities facing job insecurity, or intersectionality relating to healthcare workers.

Our rapid systematic review of reviews lacked evidence on the effectiveness of interventions, and we can therefore not recommend what specific actions should be taken by practitioners, healthcare organisations, and governments. We also found very little evidence from previous coronavirus outbreaks such as SARS and MERS. This is a missed opportunity to learn from past events that should not be repeated. We do however conclude from our review that the following types of support should be explored as potentially fruitful avenues of mental health promotion and prevention:

- organisational support in healthcare in hospital and in the community; adequate staffing levels, clear communication and guidance, mental health training for staff, resources, and individualised support for those working from home,

- the promotion of peer support in the workplace,

- media and social media guidance to help people manage mental health impacts of a continuous news cycle with distressing narratives, and

- tangible governmental support for key workers in hospital and in the community (fair pay for healthcare workers, availability of protective equipment, staffing shortages in healthcare), which should be prioritised over and above moral support initiatives such as 'clap for carers'.

As one of our panel members emphasised, learning from the early months of the COVID-19 pandemic can help us prepare for the future, including for future pandemics. This may mitigate the negative impacts on mental health for everyone and particularly for those more likely to be affected.

## Supporting information

**S1 Appendix. PRISMA 2009 checklist.**
(DOC)

**S2 Appendix. Full search strategy.**
(DOCX)

**S3 Appendix. Systematic reviews included but not cited.**
(DOCX)

## Acknowledgments

We are grateful for the contributions of the panel of healthcare workers and to Lena Smidt for her help with literature searches.

## Author Contributions

**Conceptualization:** Eleonora P. Uphoff, Chiara Lombardo, Gordon Johnston, Lauren Weeks, Sarah Dawson, Catherine Seymour, Antonis A. Kousoulis, Rachel Churchill.

**Data curation:** Eleonora P. Uphoff, Lauren Weeks, Sarah Dawson, Catherine Seymour.

**Formal analysis:** Eleonora P. Uphoff, Lauren Weeks, Catherine Seymour.

**Methodology:** Mark Rodgers.

**Supervision:** Antonis A. Kousoulis, Rachel Churchill.

**Writing – original draft:** Eleonora P. Uphoff.

**Writing – review & editing:** Chiara Lombardo, Gordon Johnston, Lauren Weeks, Mark Rodgers, Sarah Dawson, Catherine Seymour, Antonis A. Kousoulis, Rachel Churchill.

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
