## [Decision Letter · Decision Letter 0]

20 May 2021

PONE-D-21-14122

Mental health among healthcare workers and other vulnerable groups during the COVID-19 pandemic and other coronavirus outbreaks: a rapid systematic review

PLOS ONE

Dear Dr. Rodgers,

Thank you for submitting your manuscript to PLOS ONE. After careful consideration, we feel that it has merit but does not fully meet PLOS ONE’s publication criteria as it currently stands. Therefore, we invite you to submit a revised version of the manuscript that addresses the points raised during the review process.

We look forward to receiving your revised manuscript.

Kind regards,

Ali Rostami

Academic Editor

PLOS ONE

Journal Requirements:

Reviewers' comments:

Reviewer's Responses to Questions

**Comments to the Author**

1. Is the manuscript technically sound, and do the data support the conclusions?

Reviewer #1: Yes

Reviewer #2: No

2. Has the statistical analysis been performed appropriately and rigorously? 

Reviewer #1: Yes

Reviewer #2: N/A

3. Have the authors made all data underlying the findings in their manuscript fully available?

Reviewer #1: Yes

Reviewer #2: Yes

4. Is the manuscript presented in an intelligible fashion and written in standard English?

Reviewer #1: No

Reviewer #2: Yes

5. Review Comments to the Author

Reviewer #1: Signs and symptoms are not listed separately for the disease.

Has there been substance abuse in patients?in any articles?

What are the study exclusion criteria?

It would be better to compare the results of studies of different ethnicities.

What is the most important finding of this study?The result of study that shows the emotional support of people involved with Covid ,has been obvious before.

What were the differences between ages groups in the type of mental disorders?

The findings of the article are generalized and not compared to the required extent. Most of the data was not used. Not all topics are explained in the same way.

There are not mentioned to the limitations of the articles, such as medical history of the subjects, family disputes, economic status, etc.

Providing a well-organized and complete view of the work done on a research topic is not summarized.

introduction did not mentioned the ignorance of the subject or its problems and complications.

There is relationship between the results and discussion, and the reasons for the differences in mental disorders were not mentioned in different articles.

Reviewer #2: This is interesting topic. However, there are some problems should be address well. I mentioned some issues as below:

Introduction:

-The introduction needs a major revision and improvement.

-Introduction should be focus on previous review studies.

-Why did you conduct a systematic review of reviews?

-Is there any previous review in this topic?

Methods:

-Why did you select two groups of different population of mental problems (adults and children)?

- Why did you select the reviews of five different population including healthcare workers, children and adolescents, patients with pre-existing conditions, and homeless?

- The mental problems are different in adults and children. Also, risk of affected with Coronavirus was different between adult and children. How did you select children and adult as population study?

Also, healthcare workers were as high-risk while the children were low- risk group.

Results

-Please discuss how did not this systematic review of reviews lead to meta-analysis?

Discussion

The study included five population studies, but the discussion and conclusion focused on one population (healthcare workers). How did the authors miss children, homeless from the study?

6. PLOS authors have the option to publish the peer review history of their article (what does this mean?). If published, this will include your full peer review and any attached files.

Reviewer #1: No

Reviewer #2: No

---

## [Author Response · Author response to Decision Letter 0]

21 Jun 2021

Reviewer #1

1 Signs and symptoms are not listed separately for the disease.

Thank you for this suggestion. On page 6, we have added that we included studies on ‘any signs or symptoms’ of the eligible mental health conditions.

2 Has there been substance abuse in patients in any articles?

We only found one mention of substance abuse linked to the COVID 19 pandemic, which is mentioned on page 13: 

“Reports of mental health symptoms in college students included anxiety, depression, substance abuse, sleeping disorders, and eating disorders [21].”

3 What are the study exclusion criteria?

The reviewer is right that we mostly state our inclusion criteria; studies which did not meet these criteria were not included. In addition, as stated on page 7: 

“Abstracts and titles for which no full-text manuscript could be obtained within two weeks of starting data extraction were excluded.”

4 It would be better to compare the results of studies of different ethnicities

We agree that there would be merit in exploring differences in results by ethnicity. Unfortunately, we were unable to do this as included reviews mostly considered a range of ethnic groups, without presenting results separately by ethnicity.

5 What is the most important finding of this study?

The result of study that shows the emotional support of people involved with Covid ,has been obvious before. We hope our abstract states the most important findings. Firstly, it is important to realise that evidence was lacking for many vulnerable groups and for interventions. Secondly, we highlight risk and protective factors, although mostly for healthcare workers, which could be used in the development of interventions.

6 What were the differences between ages groups in the type of mental disorders?

Because we report on findings from systematic reviews, we were restricted by the presentation of results in those reviews. We were therefore unable to extract data on differences between age groups. 

We do present findings for children and for adults separately, but reviews were too different in their focus, design and methods to be able to make meaningful comparisons. 

7 The findings of the article are generalized and not compared to the required extent. Most of the data was not used. Not all topics are explained in the same way

Unfortunately, the format of our ‘review of reviews’ meant that we did not have access to data from individual studies. Also, we present more evidence (and discussion) on healthcare workers because this is what we found in the literature. We completely agree with the reviewer that more evidence on specific population groups is needed.

8 There are not mentioned to the limitations of the articles, such as medical history of the subjects, family disputes, economic status, etc.

We agree with the reviewer that these factors could influence the results and are important to consider where possible. Due to the nature of our ‘review of reviews’ we were unable to assess individual studies within the included reviews. However, we used the CASP checklist to assess the quality of included reviews. 

9 Providing a well-organized and complete view of the work done on a research topic is not summarized.

We are not entirely sure what the reviewer is referring to. Perhaps they were wondering whether other reviews of reviews exist on this topic. To clarify, we have added a statement on page 5: “At the time of conducting our study, we were not aware of another review of systematic reviews on this topic.”

See also response to comment #12.

10 introduction did not mentioned the ignorance of the subject or its problems and complications.

We hope to have addressed the importance of the topic in the following ways:

- The first paragraph highlights the enormous burden of disease and deaths from COVID-19.

- The third paragraph discusses the problems vulnerable groups may encounter.

If required, could the editor please advise what other information may be needed?

11 There is relationship between the results and discussion, and the reasons for the differences in mental disorders were not mentioned in different articles.

We agree with the reviewer that it is interesting to note the differences in prevalence rates between mental disorders. However, as the included reviews were heterogenous in terms of aims, methods, and selection criteria, we cannot explain nor speculate on reasons for these differences. We hope future research will be able to provide these answers. 

 Reviewer #2

12 Introduction:

-The introduction needs a major revision and improvement.

-Introduction should be focus on previous review studies.

-Why did you conduct a systematic review of reviews?

-Is there any previous review in this topic? Thank you for your observations.

As stated on page 5, “We used rapid review methodology based on guidance from the Cochrane Rapid Review Methods Group, to ensure a pragmatic yet high quality systematic review.” 

We did not identify other reviews of reviews on this topic. The purpose of our reviews of reviews was to identify previously published systematic reviews; we therefore did not discuss these reviews as part of the introduction.

To clarify, we have added a statement on page 5: “At the time of conducting our study, we were not aware of another review of systematic reviews on this topic.”

13 -Why did you select two groups of different population of mental problems (adults and children)?

We did not select these groups a priori; our results reflect the literature we identified. Since exposure to the pandemic and mental health problems may differ between children and adults, we present results separately for these groups.

14 Why did you select the reviews of five different population including healthcare workers, children and adolescents, patients with pre-existing conditions, and homeless? We agree with the reviewer that other groups are important. However, we did not select these five populations; unfortunately, we did not find any review evidence for other groups which fitted our selection criteria. On page 20 of the discussion section, we therefore state: 

“Evidence was lacking for vulnerable groups other than healthcare workers.”

15 The mental problems are different in adults and children. Also, risk of affected with Coronavirus was different between adult and children. How did you select children and adult as population study? See response to comment no. 13.

16 Also, healthcare workers were as high-risk while the children were low- risk group.

 To guide us as to the perceived vulnerability of specific groups, we used the paper by Holmes et al (ref 1). We agree that today, the latest available evidence may make a case for children being a vulnerable group too.

17 Results Please discuss how did not this systematic review of reviews lead to meta-analysis?

As stated on page 8, a meta-analysis was not deemed appropriate for this study (a priori): 

“The extracted data were used to produce a narrative synthesis of results by type of inequality group, as we expected the study designs and data to be too heterogeneous for meta-analyses to be appropriate.”

18 Discussion The study included five population studies, but the discussion and conclusion focused on one population (healthcare workers). How did the authors miss children, homeless from the study?

The ‘summary of findings’ on page 19 briefly reports on the results on all population groups we encountered. Since most of the evidence we found was on healthcare workers, a panel of healthcare workers was asked to help us interpret findings. The discussion summarises their input, as well as pointing out that little evidence was available for other population groups.

In the ‘Implications for research and practice’ section we call for future research with “attention for those belonging to multiple vulnerable or minority groups, such as older people with physical health problems, ethnic minorities facing job insecurity, or intersectionality relating to healthcare workers.”

---

## [Decision Letter · Decision Letter 1]

5 Jul 2021

Mental health among healthcare workers and other vulnerable groups during the COVID-19 pandemic and other coronavirus outbreaks: a rapid systematic review

PONE-D-21-14122R1

Dear Dr. Rodgers,

We’re pleased to inform you that your manuscript has been judged scientifically suitable for publication and will be formally accepted for publication once it meets all outstanding technical requirements.

Kind regards,

Ali Rostami

Academic Editor

PLOS ONE

Additional Editor Comments (optional):

Reviewers' comments:

Reviewer's Responses to Questions

**Comments to the Author**

1. If the authors have adequately addressed your comments raised in a previous round of review and you feel that this manuscript is now acceptable for publication, you may indicate that here to bypass the “Comments to the Author” section, enter your conflict of interest statement in the “Confidential to Editor” section, and submit your "Accept" recommendation.

Reviewer #1: (No Response)

Reviewer #2: All comments have been addressed

2. Is the manuscript technically sound, and do the data support the conclusions?

Reviewer #1: (No Response)

Reviewer #2: Yes

3. Has the statistical analysis been performed appropriately and rigorously? 

Reviewer #1: (No Response)

Reviewer #2: Yes

4. Have the authors made all data underlying the findings in their manuscript fully available?

Reviewer #1: (No Response)

Reviewer #2: Yes

5. Is the manuscript presented in an intelligible fashion and written in standard English?

Reviewer #1: (No Response)

Reviewer #2: Yes

6. Review Comments to the Author

Reviewer #1: (No Response)

Reviewer #2: Dear Authors

I think that you improved well the revised manuscript.

I thank you for sufficient response/correction.

7. PLOS authors have the option to publish the peer review history of their article (what does this mean?). If published, this will include your full peer review and any attached files.

Reviewer #1: No

Reviewer #2: No

---

## [Editor Report · Acceptance letter]

13 Jul 2021

PONE-D-21-14122R1 

Mental health among healthcare workers and other vulnerable groups during the COVID-19 pandemic and other coronavirus outbreaks: a rapid systematic review 

Dear Dr. Rodgers:

I'm pleased to inform you that your manuscript has been deemed suitable for publication in PLOS ONE. Congratulations! Your manuscript is now with our production department. 

Kind regards, 

on behalf of

Dr. Ali Rostami 

Academic Editor

PLOS ONE